# Non-Structural Carbohydrates Accumulation in Seedlings Improved Flowering Quality of Tree Peony under Forcing Culture Conditions, with Roots Playing a Crucial Role

**DOI:** 10.3390/plants13202837

**Published:** 2024-10-10

**Authors:** Shuaiying Shi, Tian Shi, Shuang Zhou, Shuangcheng Gao, Yuan Zhao, Guoan Shi

**Affiliations:** 1Henan Comprehensive Utilization Engineering Technical Research Center for Peony, College of Mudan, Henan University of Science and Technology, Luoyang 471023, China; shishuaiying789@163.com (S.S.); charismatic0125@163.com (T.S.); zhoushuang2001@163.com (S.Z.); gsczml@163.com (S.G.); zhaoyuan17@yeah.net (Y.Z.); 2College of Landscape Architecture and Art, Henan Agricultural University, Zhengzhou 450002, China

**Keywords:** *Paeonia suffruticosa*, non-structural carbohydrates, remobilization, flowering period, potted seedling, flowering index

## Abstract

(1) Tree peony (*Paeonia suffruticosa* Andrews) is a woody ornamental plant originating from China, and beloved by people worldwide. Non-structural carbohydrates (NSCs) play a crucial role in regulating the flowering quality of tree peonies in both field and potted conditions. However, the effects of NSCs accumulation and allocation in various organs during the vegetative growth stage on the flowering quality of tree peony under forcing culture remains unclear. (2) Two-year-old grafted seedlings of tree peony cv. ‘Luoyanghong’ were subjected to orthogonal treatments to investigate the role of NSCs accumulation in plants’ developmental process. We measured leaf photosynthetic capacity, NSCs accumulation in the organs of seedlings, observed key ornamental characteristics of flowering quality under forcing culture conditions, and evaluated the qualities of seedlings and flowers using the seedling index (SI) and flowering index (FI), respectively. (3) There was a significant positive correlation between leaf photosynthetic capacity and NSCs accumulation in both the whole plant and roots of potted tree peony. Roots were identified as the primary organs for NSCs accumulation in potted tree peonies. Sufficient NSCs accumulation in the plant, particularly in the roots during the defoliation period, was essential not only for enhancing the seedling quality of potted tree peonies but also for improving the flowering quality under forcing culture conditions. Both the seedling index (SI) and flowering index (FI) exhibited a significant dose-response with increasing root NSCs accumulation at defoliation. The T3 group, which involved slight root pruning (by 25%), combined with a high-concentration rooting agent (750 mg·L^−1^) and *Metarhizium anisopliae* (20 million U·mL^−1^), resulted in the highest photosynthetic capacity, SI, FI and NSCs accumulation status (NSCAR), making it the optimal treatment combination. (4) This finding indicates that increasing NSCs accumulation in the roots of potted tree peonies is a crucial biological foundation for producing high-quality potted flowers under forcing culture conditions, which provide new insights into the importance of NSCs in tree peony flowering and may improve the production technology for high-quality potted tree peony flowers under forcing culture conditions.

## 1. Introduction

Tree peonies, known as *Paeonia suffruticosa* Andr., originated from China and boast a heritage stretching over 1500 years. Renowned for their brilliant and fragrant flower, they are extolled as the ‘king of flowers’, symbolizing wealth and prosperity in Chinese culture [1]. Historically, field planting has been the predominant cultivation method for tree peonies due to its low costs and ease of management [2]. However, the natural flowering period of field-planted tree peonies is short and concentrated, posing the challenge of extending the blooming period to meet tourist demands [3]. Interestingly, potted tree peonies play an increasingly important role in off-site and off-season industrial production due to their mobility and ease in controlled environments [4,5]. Significant improvements in the flowering quality of potted tree peony have been achieved through the efforts of horticulturists [6,7]. Some researchers have proposed that flowering of tree peony requires significant nutrient consumption [8]. However, there are few reports detailing the role of nutrient accumulation in the flowering process of tree peony.

Tree peony, a perennial deciduous shrub, stores photosynthetic assimilates in stems and roots as material and energy sources for organ reconstruction in the subsequent growth period [9]. The growth of tree peony buds and leaves is almost simultaneous from spring to early summer, leading to intense competition for nutrient remobilization between vegetative and reproductive organs. This nutrient competition often results in bud abortion [10]. Defoliation, which reduces leaf competition for nutrients, is beneficial for increasing the flowering quality under forcing culture conditions [11].

Roots are one of the main nutrient providers for the growth of buds and leaves. The concentration of carbon (C), nitrogen (N), phosphorus (P), and potassium (K) in the roots significantly decreases after flowering [12], indicating that enhancing the nutrient mobilization capacity in roots may improve the flowering quality of potted tree peony under forcing culture conditions [13]. Inorganic elements (N, P, and K) for plant growth and development can be quickly supplemented through irrigation [14]. However, off-season potted tree peony production faces certain challenges and difficulties in the reconstruction of tissue carbon skeleton during the plant development process. In fact, the storage of organic carbon of roots during the vigorous growth period is a key factor for the new organ reconstruction of tree peony in the next growth period.

Organic carbon in plants includes structural carbohydrates (SCs) and non-structural carbohydrates (NSCs). SCs are polysaccharides used to build and solidify plant tissues, including cell walls, stems and stalks, making them difficult to remobilize [15,16]. In contrast, NSCs are the main form of carbon storage and can undergo frequent transformations, providing the building blocks for plant reconstruction and versatile resources for metabolic processes [17].

NSCs, which consist mainly of soluble sugars and starch, are crucial substrates for construction and energy, and even act as signaling triggers in maintaining plant life and defense [18,19]. In adverse conditions, such as low temperatures and drought, when photosynthates cannot meet the demands of metabolism, storage NSCs are remobilized to help plants survive unfavorable conditions [20]. Due to morphological and physiological diversity, plants have evolved different NSCs storage and mobilization strategies [21]. Deciduous broad-leaved trees stored more NSCs in their roots than evergreen species. When low temperatures occur, stored starch in roots converts to soluble sugars, which protect plants from low-temperature damage and provide energy for leave and stem growth in spring [22]. NSCs accumulation during the previous growing season also significantly impacts deciduous plant regrowth in the next season [23]. Our previous research found that the growth status of above-ground parts of the tree peony is closely related to the accumulation of NSCs [24]. Major ornamental characteristics of tree peony, such as flowering rate, petal color, flower diameter and flowering period, are significantly influenced by NSCs content [25,26,27]. However, our current understanding of NSCs’ impact on tree peony flowering is limited. Previous studies primarily focused on the influence of NSCs concentration on growth during the normal season, rarely addressing the accumulation of NSCs and its effect on off-season flowering quality under forcing culture conditions.

This study aimed to investigate the specific role of NSCs accumulation in regulating the flowering quality of tree peony under forcing culture conditions. An orthogonal test was designed with three factors: root pruning, rooting agent and *Metarhizium anisopliae*. The objective was to explore the relationship between NSCs accumulation in organs and flowering characteristics of potted tree peony. This experiment may provide valuable technical support for the simplified and high-quality production of potted off-season tree peony flowers under forcing culture conditions.

## 2. Results

### 2.1. Changes in the Photosynthetic Capacity of Leaves

Leaves are the main organs for NSCs production in potted tree peony plants, and high photosynthetic capacity is essential for accumulating sufficient NSCs. These NSCs play important roles in supporting seedling development and organ reconstruction for the next growth stage. Photosynthesis parameters were measured by the Li-6400 system in May 2021. The results showed no significant difference in *P*n among all groups (Table 1). However, groups with slight root pruning showed lower *G*s, *T*r, and *C*i, and higher WUE, indicating that a well-developed root system is beneficial for improving the efficiency of water and CO_2_ utilization in the leaves of potted tree peony.

Leaf area per plant (LA) is also a key factor influencing the photosynthetic assimilation capacity of seedlings. The results indicated that LA decreased with increased root pruning and showed an increasing trend with increased concentrations of rooting agent. This suggests that root pruning and rooting agent have different effects on leaf development: root pruning reduces nutrient storage in roots, inhibiting leaf development, while rooting agent can alleviate these negative effects.

Antioxidant capacity plays an important role in the continuous synthesis of NSCs in leaves. The results showed that leaf POD activity of slight root pruning groups was higher than in the control. Concurrently, the content of MDA, a product of membrane lipid peroxidation, decreased with increasing rooting agent concentration. These findings indicate that root pruning and rooting agent have different effects on antioxidant capacity in potted tree peony leaves. Root pruning may reduce the storage nutrient in roots, thereby limiting leaf expansion, while rooting agent can mitigate this negative effect.

### 2.2. Differences in Morphology and NSCs accumulation of Potted Seedlings

The morphological characteristics and NSCs accumulation status of potted tree peony seedlings were measured after defoliation in October 2021, revealing significant differences among the 10 groups. Abundant fibrous roots developed at the coarse roots in all groups, an important adaptive phenomenon to the potted environment (Appendix A). Except for T1 and T6, the stem length (SL) in all groups was significantly shorter than that of T0, indicating that root pruning showed a noticeable dwarfing effect (Table 2). Meanwhile, stem diameter (SD), whole plant biomass (WPB), NSCs accumulation in the root (NSCAR), NSCs accumulation in the total plant (NSCAT), and seedling index (SI) decreased with increasing root pruning but showed an upward trend with higher concentrations of the rooting agent. Notably, over 96% of NSCs were stored in the root in all groups. The WPB and NSCAT of T2, T3, and T6 exceeded those of T0. The growth quality of potted tree peony seedlings, evaluated by SI, indicated that the slight and moderate root pruning groups achieved or surpassed the quality of T0, with T3 exhibiting the highest SI.

### 2.3. Differences in Flowering Quality under Forcing Culture Conditions

Under forcing culture management, the flowering characteristics among different groups exhibited notable differences (Appendix A; Table 3). Compared to control (T0), plant height (PH) of T7 and T8 groups significantly reduced by 26.8% and 28.9%, respectively, while branches number (BN) increased by approximately 27%, 60%, 60%, 94%, 110%, 94%, 83%, 94% and 38%, respectively (Table 3). Severe root pruning (T7 and T8) resulted in crown width (CW), leaf length and SPAD values that were markedly lower than those of T0. However, there was no significant difference in the branch diameter (BD) and leaf thickness (LT). The moderate root pruning groups (T4, T5 and T6) had the highest flowering number per plant (FNP), likely due to their high branching ability. There was no significant difference in flower diameter (FD) among all groups except for T7 and T8. The value *L** increased with increasing degree of root pruning, while the value *a** and anthocyanin content (PAC) decreased. Flavonoid content in petals (PFC)increased with the concentration of the rooting agent. There was no significant difference in soluble sugars’ content in petals compared to T0. The flowering quality under forcing culture conditions was further evaluated using the flowering index (FI), revealing that the FI of T2, T3, T5, and T6 was significantly higher than that of T0, with T3 achieving the highest value. This indicates that optimized transplant management measures could significantly improve the flowering quality of tree peony under forcing culture.

Furthermore, analysis of the effect of treatments on FI using range analysis showed that the influence of the three treatments on the FI ranked from high to low as follows: root pruning > rooting agent > *Metarhizium anisopliae*. The optimal levels for each factor were slight root pruning, high concentration of rooting agent, and *Metarhizium anisopliae*, respectively. The optimal group was T3: A1B3C3, which involved 25% root pruning intensity, irrigation with 750 mg·L^−1^ rooting agent, and 20 million U·mL^−1^
*Metarhizium anisopliae* (Table 4), achieving optimal flowering quality for the regulation of the opening stage of potted tree peony.

### 2.4. Correlations Analysis of Growth Parameters of Potted Seedlings and Off-Season Flower

As previously mentioned, there are differences in the growth parameters of potted seedlings and off-season flowering quality among different groups. To analyze the intrinsic relationship between these parameters, a simple correlation analysis was conducted. The results showed that photosynthesis parameters of leave significantly affect NSCs accumulation (Figure 1a). Among these parameters, *G*s and *T*r were positively and negatively correlated with NSCAT and NSCAR, respectively, while WUE, LA and POD, key indexes of photosynthetic capacity, had significant or extremely significant positive correlations with NSCAT and NSCAR. Furthermore, NSCAT and NSCAR also positively correlated with SL, SD, WPB and SI of potted seedlings. The correlation coefficient of NSCAR with SI reached 0.668, indicating that improving of photosynthetic capacity is beneficial for NSCs assimilation, and sufficient NSCs accumulation is foundational for strengthening seedling growth.

The parameters of flowering and vegetative growth of flowers are interrelated (Figure 1b). FD was positively correlated with PH, CW, SPAD, BW and LL. SPAD was positively correlated with BW, LL, FD, PAC, PFC and PSSC. Both FD and SPAD have negative correlations with BN. FI, a comprehensive evaluation index of potted tree peony flowering, showed significant positive correlations with CW, SPAD, BW, LL, FNP, FD, *a**, PAC and PFC, while it was negatively correlated with *L** and LT.

### 2.5. Correlation Network Analysis between Growth Parameter of Potted Seedling and Off-Season Flowering

The correlation network of growth parameters for potted seedlings and off-season flowering quality was constructed to identify the key factors in tree peony growth (Figure 2). The correlation threshold was set to 0.4 for positive correlations and −0.4 for negative correlations (*p* < 0.05). The nodes with red background indicated stronger correlations, signifying essential factors. The results showed that NSCAR, NSCAT and FI were key factors in tree peony growth. Therefore, we suggest that NSCs accumulation in roots of potted seedlings may play an important role in the flowering processes of forcing-cultured tree peony.

### 2.6. Dependency between SI and FI on Roots’ NSCs accumulation

At the time of defoliation, more than 96% of NSCs were stored in the roots of potted tree peony seedlings (Table 2). Therefore, the NSCs storage capacity of the total plant depends primarily on that of the roots. The dependency relationship of SI and FI on NSCs accumulation was analyzed using scatter plots, revealing that both SI and FI significantly increased with the increase in NSCAR (Figure 3). The determination coefficients between SI and NSCAR, FI and NSCAR are 0.4530 and 0.5129 (*p* < 0.01), respectively. This further confirms that NSCs accumulation played a crucial role in the growth of potted tree peonies and has significant dosage effects on the qualities of seedlings and flowering. Further analysis of the dependency of SI and FI (Figure 4) found a positive correlation between FI and SI, with a determination coefficient of 0.4825 (*p* < 0.01). Therefore, the flowering quality of forcing-cultured tree peony highly depends on the SI of potted base seedlings.

## 3. Discussion

### 3.1. Seedling Quality Depends upon NSCs accumulation of Potted Tree Peony

Under normal conditions, plants assimilate CO_2_ from the atmosphere via photosynthesis, using most of it in biomass construction and metabolism. A smaller fraction, in the form of soluble sugars or starch, is stored in NSCs pools [28]. NSCs play distinct functional roles, including matter transport, energy metabolism, osmoregulation, development, signaling regulation and defense when carbon assimilation cannot meet the demand. Therefore, NSCs accumulation is an integrative function relating to all life activities of plants [29].

Leaf being the main photosynthesis organ, its capacity for carbon uptake is affected by inherent and extrinsic factors, such as temperature and water in the environment, chloroplast structure, leaf morphology and antioxidant capacity [30]. Our results indicated that LA, WUE and POD were key indicators of the photosynthetic capacity of leaves, which positively correlated to NSCs accumulation and is significantly affected by root pruning degree. This suggested that NSCs accumulation influences leaf photosynthesis interdependently. For example, root pruning reduces NSCs storage in roots, which inhibits the photosynthetic capacity of leaves during vigorous growth periods, resulting in a decrease in NSCs accumulation. Interestingly, rooting agent treatment could promote leaf growth and improve WUE, which is beneficial for NSCs accumulation. This may be related to the regulatory effect of phytohormones in rooting agent on leaves [31].

In their long-term evolution, plants have developed one or more NSCs pools in their bodies to avoid the risk of NSCs depletion, providing energy for various life activities [32,33]. NSCs are distributed and stored in various organs, enabling plants to balance “growth” and “resistance” [21]. Our results showed that over 96% of NSCs in tree peony seedlings were stored in roots after defoliation, indicating that roots are the main NSCs pools. Fewer NSCs accumulated in stems, serving as auxiliary NSCs pools. However, this distribution may change as the plant ages and during growth seasons.

The natural living environments of plants are variable, so plants, and long-lived trees, in particular, are likely to face stressful conditions throughout their lives. Adversities such as drought, high temperature, low temperature, excessive shading, biotic and abiotic stress can temporarily or permanently destroy plant photosynthetic organs, which limits carbon uptake and forces plants to rely on NSCs storage [34,35,36]. Consequently, a plant’s ability to remobilize and reuse stored NSCs for metabolic activity, defense, and osmoregulation is directly linked to its survival [37,38,39,40]. Kabeya found that the seasonal distribution of NSCs is closely related to plant nutrient status in high-elevation plants, suggesting that NSCs can serve as indicators for assessing plants’ nutrient status [41]. The decreased NSCs concentration in *P. sylvestris* seedlings due to prolonged storage negatively affects their survival and growth after planting [42].

Compared to field cultivation, potted plants live in a small, and unstable space with restricted root growth, making them more susceptible to biotic and abiotic stress. Therefore, NSCs reserves in potted seedlings are considered a key quality attribute that can improve seedling field performance in out-planting [43]. Our result showed that NSCAR and NSCAT had significant correlations with SL, SD, WPB and SI of potted seedlings, indicating that NSCs are essential building blocks for biomass (Figure 1a). Since more than 96% of NSCs are stored in the roots of potted tree peonies, NSCAR played a more important role in tree peony seedling growth than NSCAT (Table 2), making it a powerful physiological index for seedling quality evaluation. SI, an effective morphological indicator, is widely used in the evaluation of seedling quality [44]. Our results showed that SI significantly and positively correlated with NSCAR, with a significant dose effect (Figure 3). This makes it useful for the rapid assessment of tree peony seedling quality in production. In summary, the T3 group had the highest photosynthetic capacity, SI and NSCAR, making it the optimal group.

### 3.2. NSCs Are Matter and Energy Sources and Signaling Triggers of Off-Season Tree Peony

NSCs are reserved not only as organic matter and energy substrates for organ reconstruction, but also as sugar signaling triggers in coordinating the vegetative and reproductive growth of plants. A study on Mediterranean oaks found that, compared with evergreen Mediterranean oaks, deciduous species stored more NSCs for new branch growth, with their biomass being significantly influenced by NSCs in early spring [45]. In sweet orange trees, flowering is the most energy-demanding phenological stage, with about 80% of the total NSCs consumed before the end of fruit drop, primarily until flowering, and the demand for NSCs increased with increasing flower number [46]. In the production of cut peony flowering, NSCs were crucial for improving flower quality, and the depletion of NSCs significantly shortened the flowering periods [47]. In the flowering process of tree peony, NSCs not only provide essential energy but also play a role in sugar signaling. T6P, a type of NSCs, regulated the expression of sugar signaling-related genes such as *PsTPS*, *PsSnRK*1 and *PsHXK*, key factors in regulating tree peony flowering [48].

FI was significantly and positively correlated with most ornamental characteristics, such as CW, SPAD, BW, LL, FNP, FD, *a**, PAC and PFC, making it the most important index for evaluating tree peony flowering quality. The determination coefficients between FI and both NSCAR and SI were 0.5129 and 0.4825, respectively, with significant dose effects. This suggested that NSCAR may be the physiological factor linking SI and FI, indicating that we could select the high-quality quality seedling for off-season potted tree peony production based on NSCs accumulation in root or SI. However, SI, being a non-destructive measure, is more suitable for practical production requirements. In our test, the T3 group, which had the highest SI, FI, and NSCs accumulation in roots, proved to be the optimal treatment.

### 3.3. Roots NSCs Remobilization Is a Milestone in Forcing Culture Conditions

Storage pools in plants are defined as stored resources that have been accumulated and can be remobilized; this mobilization may occur daily, seasonally or decadally in different organs [49]. The remobilization of accumulated NSCs varies across organs or years. For example, during the flowering of sweet oranges, the root serves as the main NSCs pool, supporting 73% of remobilized NSCs for the entire plant [44]. Under normal conditions, NSCs accumulated during the growing season are unlikely to be fully depleted, with previously stored NSCs being preserved in deeper structures as old NSCs [50]. Compared to old NSCs, the new NSCs have a high turnover rate, suggesting that plants prefer to use newly accumulated NSCs over old ones [51].

For tree peony, increased remobilization of NSCs through the application of GA can promote flowering and improve flower quality [52]. In our study, the accumulated NSCs in roots played a more significant role in tree peony flowering than those in stem (Figure 2). The FI increased with NSCAR rapidly at first and then slowly, exhibiting significant dosage effects. This suggested that remobilized NSCs in seedlings likely meet the need for flowering in optimal groups. However, the quantification of remobilized NSCs in different organs during tree peony flowering requires further experimentation. Additionally, investigating whether increasing the remobilization of NSCs, primarily by phytohormone, temperature, gas and water [30,53], is beneficial in enhancing tree peony flowering quality is a promising area for exploring.

## 4. Materials and Methods

### 4.1. Experimental Site and Design

Two-year-old grafted plants of *Paeonia suffruticosa* (‘Luoyanghong’) were cultivated at the potted tree peony research station of Henan University of Science and Technology, located in Luoyang of Henan province, China (Latitude 34°47′18″ N, and longitude 112°34′40″ E, elevation 191 m). This experiment utilized an orthogonal design with L_9_(3^4^) orthogonal tables. Four hundred robust, disease-free, and uniformly grafted seedlings were selected in October 2020. These plants were randomly divided into 10 groups, and coarse roots were pruned to different degrees (Treatment A: 25%, 33%, and 50% of total root length removed). Subsequently, the plants were potted in black plastic pots (25 cm height, 22 cm inner diameter) and irrigated with different concentrations of rooting agents (Treatment B: 250, 500, and 750 mg·L^−1^) and *Metarhizium anisopliae* suspension (Treatment C: 10, 15, and 20 million U·mL^−1^, Greenation, Chongqing, China). The treatments are detailed in Table 5.

### 4.2. Potted Seedling Growth and Flowering Regulation

The potting substrate was primarily composed of peat and high-temperature decomposed organic fertilizer, with a bulk density of 0.3 g·cm^−3^ and a pH of 6.7. The substrate contained 400 mg·g^−1^ of organic matter, 230 mg·g^−1^ of total carbon, 15 mg·g^−1^ of total nitrogen, 500 μg·g^−1^ of available phosphorus, and 30 μg·g^−1^ of available potassium. Each pot was filled with approximately 5.0 kg of substrate with a 50% moisture content. These potted tree peonies were placed under a shade net with a 40% shading rate and received normal water and fertilizer management until defoliation in October 2021. After defoliation in autumn 2021, the biomass of each organ of the potted seedlings was measured, with four repetitions. Subsequently, four other potted tree peonies were moved into −2 °C cold storage for 35 days of vernalization treatment, and then transported to a greenhouse to regulate the flowering period under forcing culture conditions until the plants opened.

### 4.3. Measurement of Photosynthesis Performance

The net photosynthetic rate (*P*n), intercellular CO_2_ concentration (*C*i), stomatal conductance (*G*s), and transpiration rate (*T*r) were determined using a Li-6400 portable photosynthesis system (Li-COR, Tucson, AZ, USA) at 25 °C, 1000 µmol m^−2^ s^−1^ light intensity. Water use efficiency (WUE) was calculated as WUE = *P*n/*T*r. Three top leaves in a plant were measured, with five repetitions per group, during the vigorous growth stage of tree peonies in May 2021. Leaf area per plant (LA) was determined using Digimizer (https://www.digimizer.com/, 25 May 2021) image analysis software (MedCalc Software Ltd., Ostend, Belgium).

### 4.4. Determination of Antioxidant Performance

Peroxidase (POD) activity was assessed using the guaiacol method [54]. Malondialdehyde (MDA) content was measured via the thio-barbituric acid method [55] during the vigorous growth stage of tree peonies in May 2021. Leaves from four plants were combined and ground into powder in liquid nitrogen, and four replicates were collected.

### 4.5. Seedlings’ Morphological Observation at Defoliation

After defoliation in October 2021, the substrate around the roots was washed with tap water. Four tree peony seedlings per group were separated into four organs (stems, old roots, woody new roots, and fine roots). Stem length (SL) and stem diameter (SD) were measured using a vernier caliper with an accuracy of 0.01 mm. The biomass of the whole plant (WPB) and each organ was measured using an electronic balance with an accuracy of 0.01 g. The robustness of the potted seedlings was evaluated based on the seedling index (SI) [56].

SI was calculated using the following formula:SI = SD/SL × WPB (1)

### 4.6. Characteristics’ Observation of Off-Season Flowering

During peak fluorescence, the ornamental characteristics of potted tree peonies were observed in each group. Branches number (BN) and flower number per plant (FNP) were recorded. Plant height (PH), leaf length (LL), and crown width (CW) were measured using a centimeter scale, and the crown width index (CWI). Flower diameter (FD) and branch diameter were measured using a reading vernier caliper, while leaf thickness (LT) was determined using a spiral micrometer. The SPAD value of leaves (SPAD) was measured with a CCM200 chlorophyll meter [57], and petal color (values *a** and *L**) was measured using an NR20XE precision color difference meter [58].

CWI was calculated using the following formula:CWI = CW/PH (2)

### 4.7. Determination of Non-Structural Carbohydrates

Each organ of four plants was collected and ground into powder in liquid nitrogen. Non-structural carbohydrates (NSCs) in different organs were measured using the anthrone colorimetric method [59]. NSCs accumulation (NSCA) was calculated according to Formula (3). The NSCs accumulation in total plant (NSCAT) was the sum of NSCA in each organ.

NSCA was calculated using the following formula.
NSCA = NSCs content × organ biomass (3)

### 4.8. Determination of Anthocyanin and Flavonoid in Petals

Petal of flowers were ground into powder in liquid nitrogen. The anthocyanin and flavonoid in petals were extracted by acidic methanol solution (with 0.1% hydrochloric acid) for 24 h. The concentration of anthocyanins was determined by the absorbance at 530 and 657 nm [60]. The content of flavonoids was determined using the Al(NO_3_)_3_-NaNO_2_ colorimetry method [61].

### 4.9. Evaluation of Off-Season Flowering Quality

Based on the national standard for potted tree peony flower classification [62], six key ornamental characteristics were divided into five levels. The flowering index (FI) was then used to comprehensively evaluate flower quality. The scoring criteria are shown in Table 6. FI was calculated according to Formula (4).
FI = *a** + FD + FNP + SPAD + BN + CI (4)

### 4.10. Statistical Analysis

The experimental data were analyzed using Microsoft Excel 2019 (Microsoft Crop., Redmond, WA, USA) and SPSS 17.0 (SPSS Inc., Chicago, IL, USA) software. Differences in the data were analyzed using one-way ANOVA with post hoc Duncan’s multiple range tests, with *p*-values less than 0.05 considered statistically significant. The creation of scatter plots of SI and FI with NSCAT were used Excel 2019. The correlation network was performed using the OmicStudio tools at https://www.omicstudio.cn/tool (accessed on 25 July 2024). Photos of potted tree peony seedlings and off-season flowers were processed using Adobe Photoshop CS6 (Adobe Systems Inc., San Jose, CA, USA).

## 5. Conclusions

In summary, this study demonstrated that high photosynthetic capacity is the base of increasing NSCs assimilation in root; moreover, sufficient NSCs are beneficial in achieving the highest seedling and flowering quality in potted tree peony. T3 group treated by light root pruning combined with high-concentration rooting agent and *Metarhizium anisopliae* have the highest photosynthetic capacity, status of NSCs accumulation, seedling and flowering quality, and this is the optimal treatment. These findings provide a new perspective in understanding the key roles of NSCs in tree peony growth and flowering and offer a measure for optimizing the cultivation practices of tree peony. This has great application value in the industrial production of off-season potted tree peony flowers. Although NSCs accumulation in plants is the endogenous material basis for improving the flowering quality of off-season tree peony, further studies on the mechanisms of NSCs remobilization in plants will provide new insights for the off-season cultivation of tree peonies.

## Figures and Tables

**Figure 1 plants-13-02837-f001:**
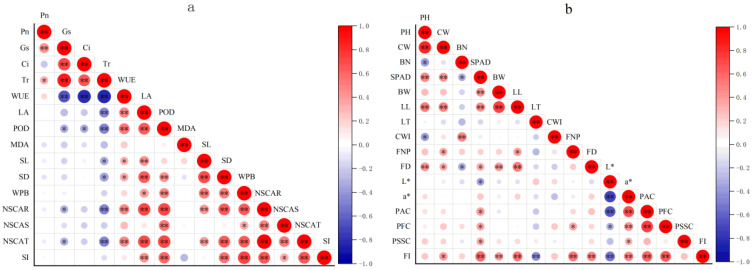
Correlation of growth parameters of potted seedlings (**a**) and off-season flowering (**b**) of tree peony. The asterisks indicate significant relation * *p* < 0.05, ** *p* < 0.01.

**Figure 2 plants-13-02837-f002:**
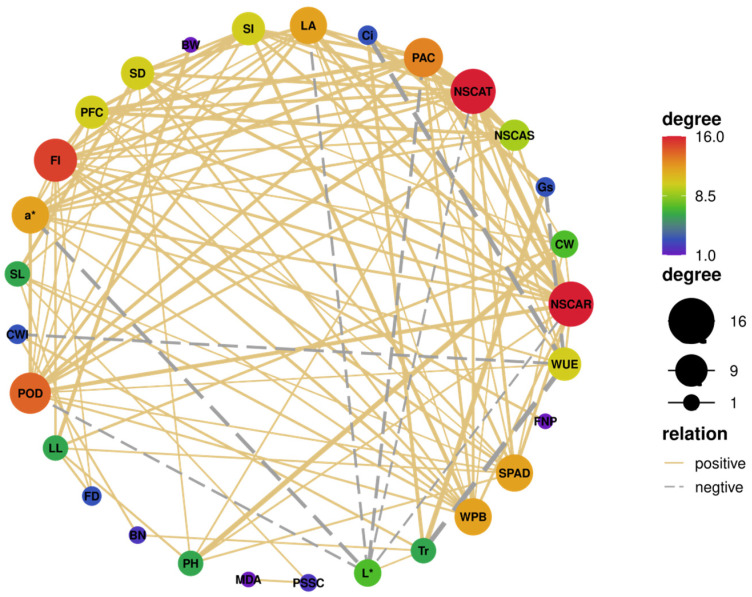
Correlation network analysis of growth parameters of seedlings and flowers of potted tree peony. The nodes with red background meant stronger correlations. The yellow solid and gray dashed lines connecting nodes mean positive and negative correlations. The area of the point is consistent with its degree of weight. The thickness of lines represents Pearson’s coefficient.

**Figure 3 plants-13-02837-f003:**
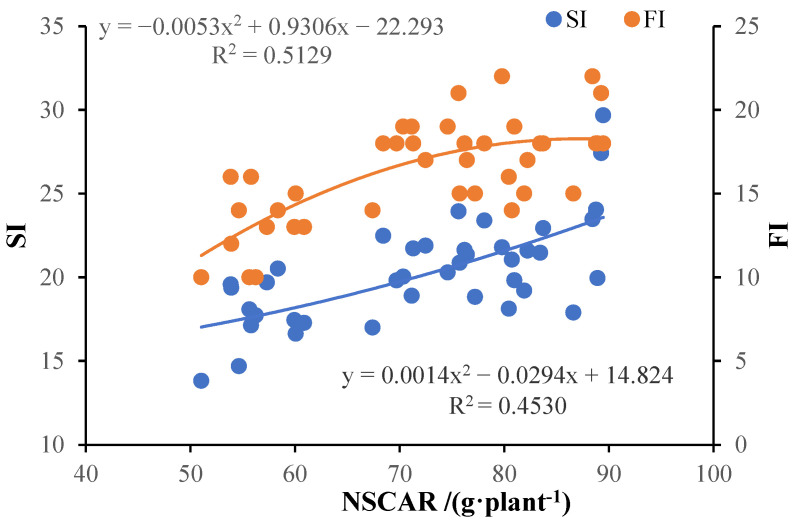
Dependence of SI of potted seedling at defoliation and FI of off-season potted flowers on NSCAT of tree peony. Blue spots and line are SI. Orange spots and line are FI.

**Figure 4 plants-13-02837-f004:**
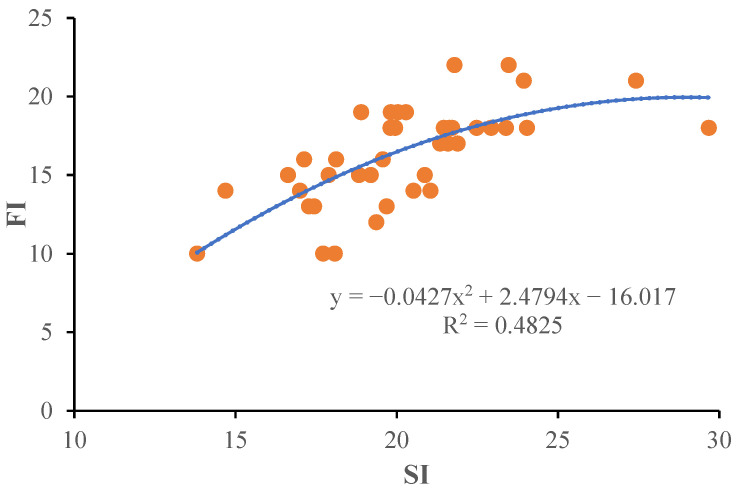
Dependence of SI of potted seedling at defoliation with FI of off-season potted flowers.

**Table 1 plants-13-02837-t001:** Changes in photosynthesis and lipid peroxidation in potted tree peony leaves.

Groups	*P*n /(µmolCO_2_ ·m^−2^·s^−1^)	*G*s /(mmolH_2_O ·m^−2^·s^−1^)	*C*i /(µmol ·mol^−1^)	*T*r /(mmolH_2_O ·m^−2^·s^−1^)	WUE /(µmolCO_2_ ·mmol^−1^ H_2_O)	LA /(cm^2^ ·plant^−1^)	POD Activity/(U·g^−1^·h^−1^)	MDA Content /(µmol·g^−1^)
T0	10.21 ± 1.51	0.09 ± 0.01 b	200.8 ± 18.0 bc	2.89 ± 0.34 de	3.54 ± 0.35 a	1725.6 ± 263.5 a	41.4 ± 13.6 c	9.79 ± 0.29 ab
T1	9.75 ± 2.23	0.09 ± 0.04 b	194.4 ± 29.1 c	2.81 ± 0.97 e	3.58 ± 0.45 a	1554.1 ± 77.6 ab	62.5 ± 11.0 b	10.07 ± 0.57 a
T2	10.81 ± 1.79	0.10 ± 0.02 ab	195.7 ± 28.5 c	3.08 ± 0.59 cde	3.57 ± 0.58 a	1345.4 ± 81.4 bc	63.7 ± 15.1 b	8.54 ± 0.37 de
T3	11.01 ± 1.65	0.10 ± 0.02 ab	198.1 ± 17.7 c	3.27 ± 0.62 bcde	3.40 ± 0.25 a	1792.3 ± 209.7 a	82.7 ± 14.2 a	8.14 ± 0.37 ef
T4	11.41 ± 1.08	0.10 ± 0.02 ab	196.2 ± 17.9 c	3.36 ± 0.46 abcd	3.42 ± 0.26 a	1269.2 ± 244.1 cd	19.8 ± 2.6 d	9.10 ± 0.33 cd
T5	10.60 ± 2.44	0.12 ± 0.03 a	225.0 ± 13.9 a	3.78 ± 0.85 ab	2.82 ± 0.23 bc	1237.0 ± 40.5 cd	27.9 ± 8.8 cd	9.28 ± 0.38 bc
T6	10.52 ± 1.05	0.11 ± 0.02 ab	222.9 ± 14.9 a	4.02 ± 0.47 a	2.63 ± 0.28 c	1434.7 ± 203.5 bc	33.5 ± 11.0 cd	8.02 ± 0.39 ef
T7	10.63 ± 1.06	0.11 ± 0.02 ab	210.7 ± 11.3 abc	3.84 ± 0.47 ab	2.78 ± 0.15 bc	1028.6 ± 112.3 d	26.4 ± 6.2 cd	10.36 ± 0.4 a
T8	10.69 ± 1.56	0.10 ± 0.02 ab	189.4 ± 18.8 c	3.53 ± 0.67 abcd	3.06 ± 0.26 b	1097.5 ± 157.0 d	15.7 ± 10.2 d	7.94 ± 0.35 ef
T9	10.64 ± 1.49	0.11 ± 0.02 ab	220.0 ± 24.3 ab	3.65 ± 0.49 abc	2.94 ± 0.43 bc	1219.2 ± 209.9 cd	19.8 ± 4.6 d	7.55 ± 0.13 f

*P*n Net photosynthetic rate, *G*s Stomatal conductance, *C*i Intercellular CO_2_ concentration, *T*r Transpiration rate, WUE Water use efficiency, LA Leaf area per plant. Note: Different letters in the same column meant significant differences at 0.05 level (mean values ± SE, n = 4).

**Table 2 plants-13-02837-t002:** Changes in morphology and NSCs accumulation of potted seedlings at defoliation.

Groups	SL /mm	SD /mm	WPB /(g·plant^−1^)	NSCAS /(g·plant^−1^)	NSCAR /(g·plant^−1^)	NSCAT /(g·plant^−1^)	SI
T0	111.6 ± 11.9 a	6.2 ± 0.1 ab	344.8 ± 39.2 a	1.42 ± 0.06 h	81.3 ± 4.3 b	82.7 ± 4.2 b	19.2 ± 1.7 bcd
T1	102.0 ± 9.0 ab	6.3 ± 0.3 ab	332.9 ± 17.3 ab	2.46 ± 0.03 b	79.1 ± 2.8 b	81.6 ± 2.8 b	20.8 ± 2.4 bc
T2	88.2 ± 4.0 bc	5.4 ± 0.1 cd	345.4 ± 39.2 a	2.92 ± 0.12 a	80.3 ± 6.2 b	83.2 ± 6.3 b	21.2 ± 2.5 bc
T3	91.4 ± 7.6 bc	6.4 ± 0.3 a	370.7 ± 15.1 a	2.01 ± 0.06 d	89.0 ± 0.5 a	91.0 ± 0.5 a	26.2 ± 2.9 a
T4	93.4 ± 14.7 bc	5.8 ± 0.7 bc	321.6 ± 32.9 ab	1.93 ± 0.08 de	71.4 ± 3.0 c	73.4 ± 3.0 c	20.2 ± 2.3 bc
T5	87.2 ± 12.0 bc	5.3 ± 0.2 d	333.3 ± 24.9 ab	1.84 ± 0.07 ef	69.9 ± 1.1 c	71.7 ± 1.2 c	20.3 ± 1.5 bc
T6	99.1 ± 12.4 ab	6.1 ± 0.4 ab	362.6 ± 24.2 a	2.33 ± 0.16 c	81.3 ± 2.8 b	83.6 ± 2.7 b	22.4 ± 0.9 b
T7	82.8 ± 5.8 c	4.7 ± 0.2 e	288.0 ± 41.1 b	1.79 ± 0.03 f	54.0 ± 2.2 d	55.7 ± 2.2 d	16.4 ± 2.6 d
T8	78.7 ± 9.7 c	4.9 ± 0.4 de	287.7 ± 39.4 b	1.65 ± 0.05 g	57.4 ± 3.1 d	59.0 ± 3.1 d	18.0 ± 2.1 d
T9	92.7 ± 4.6 b	5.9 ± 0.3 abc	292.5 ± 27.9 b	1.28 ± 0.04 i	58.1 ± 2.1 d	59.4 ± 2.1 d	18.7 ± 1.7 cd

SL Stem length, SD Stem diameter, WPB Whole plant biomass, NSCAS NSCs accumulation in stem, NSCAR NSCs accumulation in root, NSCAT NSCs accumulation in total plant. SI Seedling index. Note: Different letters in the same column meant significant differences at 0.05 level (mean values ± SE, n = 4).

**Table 3 plants-13-02837-t003:** The morphological and flowering quality differences in off-season potted tree peony flowers.

Groups	T0	T1	T2	T3	T4	T5	T6	T7	T8	T9
PH/(cm)	47.8 ± 5.4 a	44.2 ± 6.2 ab	42.0 ± 2.7 ab	41.5 ± 4.7 abc	43.8 ± 3.3 ab	45.3 ± 7.1 ab	41.8 ± 7.3 abc	35.0 ± 4.1 bc	34.0 ± 3.7 c	43.3 ± 2.1 ab
CW/(cm)	57.1 ± 4.6 ab	56.5 ± 11.2 ab	52.3 ± 5.4 ab	55.8 ± 4.3 ab	57.1 ± 9.1 ab	59.1 ± 6.0 a	60.3 ± 8.0 a	48.0 ± 4.0 bc	44.8 ± 3.7 c	56.5 ± 4.3 b
BN/(N)	1.8 ± 0.5 d	2.3 ± 0.5 cd	3.0 ± 1.1 abc	3.0 ± 0.0 abc	3.5 ± 0.6 ab	3.8 ± 1.0 a	3.5 ± 0.6 ab	3.3 ± 0.5 abc	3.5 ± 0.6 ab	2.5 ± 0.6 bcd
SPAD	38.3 ± 3.5 a	33.4 ± 2.5 bcd	35.5 ± 1.4 ab	35.3 ± 1.1 ab	34.0 ± 2.6 abc	34.5 ± 2.2 abc	35.5 ± 3.8 ab	28.8 ± 4.6 d	30.1 ± 2.3 cd	32.5 ± 3.1 bcd
BD/(mm)	8.45 ± 1.54	7.82 ± 0.59	7.60 ± 0.64	8.00 ± 0.60	8.05 ± 0.56	8.09 ± 0.56	7.61 ± 0.57	7.21 ± 0.96	7.58 ± 0.85	8.54 ± 0.47
LL/(cm)	35.5 ± 4.2 a	32.8 ± 2.6 ab	31.6 ± 2.1 ab	33.3 ± 3.1 ab	34.1 ± 3.5 a	36.5 ± 3.8 a	32.1 ± 2.2 ab	29.0 ± 3.3 b	28.3 ± 2.3 b	34.5 ± 2.7 a
LT/(mm)	0.17 ± 0.01	0.17 ± 0.01	0.16 ± 0.01	0.16 ± 0.01	0.17 ± 0.01	0.15 ± 0.01	0.16 ± 0.01	0.17 ± 0.01	0.17 ± 0.01	0.17 ± 0.01
CWI	1.20 ± 0.05 c	1.27 ± 0.10 bc	1.24 ± 0.08 bc	1.35 ± 0.06 ab	1.30 ± 0.12 bc	1.32 ± 0.09 bc	1.45 ± 0.07 a	1.38 ± 0.09 ab	1.32 ± 0.04 bc	1.31 ± 0.07 bc
FNP/(N)	1.8 ± 0.5 bc	1.3 ± 0.5 bc	1.8 ± 0.5 bc	2.3 ± 0.5 ab	2.0 ± 0.8 ab	3.0 ± 0.1 a	2.3 ± 1.0 ab	1.0 ± 0.1 c	1.8 ± 1.0 bc	1.5 ± 0.6 bc
FD/(mm)	189.2 ± 4.4 ab	198.7 ± 2.8 a	191.4 ± 12.0 ab	181.7 ± 16.2 ab	175.2 ± 20.1 abc	188.5 ± 17.8 ab	175.7 ± 21.6 abc	150.6 ± 10.5 c	156.7 ± 29.5 bc	189.7 ± 15.4 a
*L**	49.6 ± 5.3 c	50.3 ± 1.8 bc	41.1 ± 3.3 d	45.7 ± 5.7 cd	48.8 ± 6.9 cd	52.8 ± 5.9 abc	52.4 ± 8.2 abc	58.9 ± 1.9 a	52.8 ± 5.7 abc	58.1 ± 3.0 ab
*a**	25.2 ± 3.7 bc	27.8 ± 0.6 bc	32.1 ± 2.5 ab	32.1 ± 1.2 a	26.2 ± 5.0 bc	21.8 ± 7.0 cd	24.2 ± 7.9 bc	19.4 ± 5.9 cd	19.7 ± 6.6 cd	17.4 ± 2.8 d
PAC/(U·g^−1^)	24.7 ± 0.7 d	29.3 ± 0.9 c	40.6 ± 1.5 a	34.2 ± 1.9 b	28.9 ± 1.2 c	21.5 ± 0.9 e	34.0 ± 1.6 b	19.4 ± 0.2 f	19.5 ± 0.7 f	16.3 ± 0.4 g
PFC/(mg·g^−1^)	26.9 ± 1.5 d	28.4 ± 0.7 cd	32.0 ± 0.8 b	34.9 ± 2.6 a	30.5 ± 1.3 bc	29.3 ± 0.5 cd	34.9 ± 2.1 a	26.7 ± 1.0 d	27.0 ± 0.9 d	27.8 ± 1.9 d
PSSC/(mg·g^−1^)	36.3 ± 2.7 ab	36.7 ± 2.4 ab	33.2 ± 1.2 ab	32.8 ± 1.3 ab	37.4 ± 3.0 a	32.1 ± 4.8 ab	34.6 ± 2.6 ab	32.8 ± 1.2 ab	32.6 ± 5.4 ab	28.6 ± 4.4 b
FI	16.0 ± 1.4 cd	16.8 ± 1.5 cd	18.3 ± 2.5 ab	19.8 ± 2.1 a	17.0 ± 2.2 abc	18.5 ± 0.6 ab	19.0 ± 2.0 ab	11.5 ± 1.9 e	13.5 ± 2.7 de	14.0 ± 1.4 de

PH Plant high, CW Crown width, BN branch number, SPAD SPAD value of leaves, BD Branch diameter, LL Leaf length, LT Leaf thickness, CWI Crown width index, FNP Flower number per plant, FD Flower diameter, *L** value *L**, *a** value *a**, PAC Petal anthocyanin content, PFC Petal flavonoid content, PSSC Petal soluble sugars content, FI Flowering index. Note: Different letters in the same column meant significant differences at 0.05 level (mean values ± SE, n = 4).

**Table 4 plants-13-02837-t004:** Range analysis of flowering index of potted tree peony under forcing cultivation.

Groups	A/Root Pruning (%)	B/Rooting Agent (mg·L^−1^)	C/*Metarhizium anisopliae* (million U·mL^−1^)	FI
T1	25	250	10	16.8 ± 1.5
T2	25	500	15	18.3 ± 2.5
T3	25	750	20	19.8 ± 2.1
T4	33	250	15	17.0 ± 2.2
T5	33	500	20	18.5 ± 0.6
T6	33	750	10	19.0 ± 2.0
T7	50	250	20	11.5 ± 1.9
T8	50	500	10	13.5 ± 2.7
T9	50	750	15	14.0 ± 1.4
K1	18.17	14.75	16.08	
K2	17.92	16.75	16.42	
K3	13.00	17.58	16.58	
R	5.17	2.83	0.50	
Order	A > B > C			
Optimal combination	A1B3C3			

**Table 5 plants-13-02837-t005:** Experimental design of orthogonal.

Groups	A/Root Pruning (%)	B/Rooting Agent (mg·L^−1^)	C/*Metarhizium anisopliae* (million U·mL^−1^)
T0 (Control)	0 (No)	0 (No)	0 (No)
T1	25 (Slight)	250 (Low)	10 (Low)
T2	25 (Slight)	500 (Moderate)	15 (Moderate)
T3	25 (Slight)	750 (High)	20 (High)
T4	33 (Moderate)	250 (Low)	15 (Moderate)
T5	33 (Moderate)	500 (Moderate)	20 (High)
T6	33 (Moderate)	750 (High)	10 (Low)
T7	50 (Severe)	250 (Low)	20 (High)
T8	50 (Severe)	500 (Moderate)	10 (Low)
T9	50 (Severe)	750 (High)	15 (Moderate)

**Table 6 plants-13-02837-t006:** The scoring criteria for potted tree peony flowers.

Indexes	Levels				
	1	2	3	4	5
*a**	<21	21–25	25–29	29–33	>33
FD/mm	<155	155–170	170–185	185–200	>200
FNP/N.plant^−1^	1	2	3	4	5
SPAD	<30	30–32	32–34	34–36	>36
BN/N.plant^−1^	1	2	3	4	>5
CI	<120	123–130	130–140	40–150	>150

According to the national standard of potted tree peony flowers classification, we divided six key ornamental characteristics into five levels (1–5) and gave corresponding scores.

## Data Availability

Data are contained within the article and Appendix A.

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
