# Peer review of "Non-Structural Carbohydrates Accumulation in Seedlings Improved Flowering Quality of Tree Peony under Forcing Culture Conditions, with Roots Playing a Crucial Role"

_plants, 2024, doi:10.3390/plants13202837_

Round 1
Reviewer 1 Report
Comments and Suggestions for Authors
Dear authors,
This work is very well designed and a worth of publishing.
The tree peony is interesting plant and the flowering industry needs to have it. I find it popular ornamental species for special occasion. Thus in this article:
-we can see how the cultivation can be improved in potted plants
-how scientific reasearch work in application in praxis
-how exactly the production can be designed.
The literature cited is accurate, the methods are very weell described with the application of good statistical model and very well graphically presented. The results are gathered in the tables that are heavey but is the large quantity of data so no other choice is possible.
Im not working directly with NCS, so other reviewers should examine the range of each parameter tested, please.
I have not found some mistakes.
I recommend this article to be publish for the improvement of tree peony industry with forced culture conditions.....
Regards,
Zvjezdana
Author Response
Response: It is a great honor to receive your recognition for this work. I wish you a pleasant and fulfilling work experience.
Reviewer 2 Report
Comments and Suggestions for Authors
In the Introduction, the sentence: "Unlike inorganic elements (N, P, and K), which can be quickly supplemented through irrigation" has no connection with the other sentences.
Throughout the text there are many abbreviations, which make the text difficult to follow.
Correlation Network Analysis Between Growth Parameter of Potted Seedling and Off Season Flowering is not very representative, it should be better explained where the conclusion was drawn that NSCs accumulation played an important role in tree peony flowering.
In subsection 2.6, a determination coefficient below 0.500 cannot be said to show a high dependence between the studied factors
Author Response
Comment 1: “In the Introduction, the sentence: "Unlike inorganic elements (N, P, and K), which can be quickly supplemented through irrigation" has no connection with the other sentences”.
Response 1: We thanks for you review this sentence. We reorganize the sentence as: Inorganic elements (N, P, and K) for plant growth and development, which can be quickly supplemented through irrigation [14]. However, off-season potted tree peonies production face certain challenges and difficulties in the reconstruction of tissue carbon skeleton during plant development process. In fact, the storage of organic carbon of roots during the vigorous growth period is a key factor for the new organ reconstruction of tree peony in the next growth period. (page2, paragraph 3, and line 66-71)
Comment 2: “Throughout the text there are many abbreviations, which make the text difficult to follow”.
Response 2: We feel the same with your. However, the use of abbreviations is an effective way to clarify the description of tables and figures. Therefore, abbreviations must be used in the text.
Comment 3: “Correlation Network Analysis Between Growth Parameter of Potted Seedling and Off Season Flowering is not very representative, it should be better explained where the conclusion was drawn that NSCs accumulation played an important role in tree peony flowering”.
Response 3: In this experiment, we measured 31 morphological, physiological, and biochemical indexes. The purpose of “Correlation Network Analysis Between Growth Parameter of Potted Seedling and Off Season Flowering (Subsection 2.6)” is to identify the critical indicators in the growth of tree peony potted seedlings and flowers from a wide range of potential factors, the result shower that NSCAR, NSCAT and FI was identified as the important factor. While the sentence of “NSCs played an important role in tree peony flowering”(line 217-218) is primarily validated through subsequent dependency analysis. Therefore, the statement is premature and we reorganize the sentence as: Therefore, we suggests that NSCs accumulation in roots of potted seedlings may play an important role in flowering processes of forcing-cultured tree peony. (page 6 ,paragraph 3, and line 217-219).
Comment 4: “In subsection 2.6, a determination coefficient below 0.500 cannot be said to show a high dependence between the studied factors”.
Response 4: Owing to the forcing-cultured tree peony is lasting and complex management process. The off-season flower quality of potted tree peonies is influenced by various factors, including potted seedlings foundation and subsequent management measures. The management practices for off-season flowering, particularly temperature, light, water, fertilizer, and applied GA, are primarily controlled manually, which makes it difficult to achieve consistent quality. In this study, statistical analysis reveals that the determination coefficients for the SI and FI using NSCAR have reached a extremely significant level.